# Interstitial Lung Disease in Neonates: A Long Road Is Being Paved

**DOI:** 10.3390/children10060916

**Published:** 2023-05-23

**Authors:** N. Kh. Gabitova, I. N. Cherezova, Ahmed Arafat, Dinara Sadykova

**Affiliations:** 1Department of Pediatrics, School of Medicine, Kazan State Medical University, 420012 Kazan, Russia; borismk1@rambler.ru (N.K.G.); irina.cherezova@gmail.com (I.N.C.); sadykovadi@mail.ru (D.S.); 2Children’s Republican Clinical Hospital, 420012 Kazan, Russia; 3Department of Pediatrics, NICU Division, Ismailia Medical Complex, Egypt Healthcare Authority, Ismailia 41511, Egypt

**Keywords:** interstitial lung disease, fibrosing alveolitis, desquamative interstitial pneumonitis, respiratory distress

## Abstract

**Background:** Interstitial lung disease (ILD) is one of the most difficult conditions in pulmonology due to difficulties in diagnosing, classifying, and treating this condition. They require invasive approaches to diagnose (e.g., lung biopsy), non-applicable methods (e.g., lung function tests in newborns), or potentially non-accessible methods (e.g., genetic testing in not-well-equipped facilities, and several weeks are required for results to be announced). They represent a heterogeneous group of diseases in which the alveolar epithelium, parenchyma, and capillaries of the lungs are damaged, which leads to changes in the pulmonary interstitium, proliferation of connective tissue, and thickening of the alveolar-capillary membranes and alveolar septa. These changes are accompanied by impaired oxygen diffusion, progressive respiratory failure, and radiographic signs of bilateral dissemination. Although adult and child classifications for ILD have evolved over the years, classification for ILD in neonates remains a challenge. **Case presentation:** Here we discuss ILD in neonates briefly, and report two rare cases of ILD (a male white neonate, two-day-old with fibrosing alveolitis, and another male white neonate, one-day old with desquamative interstitial pneumonitis), with these diagnoses initially thought to be presented only in adulthood. Lung biopsy and histopathological findings of the two neonates have shown mononuclear cells in the alveolar spaces, and thickening of the alveolar walls confirmed the diagnosis of fibrosing alveolitis in one neonate, and desquamation of the large mononuclear cells in the intra-alveolar space in the other neonate, with the diagnosis of desquamative interstitial pneumonitis being confirmed. Interstitial lung disease lacks a consensus guideline on classification and diagnosis in neonates, rendering it one of the greatest challenges to pediatricians and neonatologists with remarkable morbidity and mortality rates. **Conclusions:** Fibrosing alveolitis and desquamative interstitial pneumonitis (DIP) are not adult-only conditions, although rare in neonates, histopathological examination and clinical practice can confirm the diagnosis. Based on our clinical practice, prenatal and maternal conditions may serve as potential risk factors for developing IDL in neonates, and further studies are needed to prove this hypothesis.

## 1. Introduction

Interstitial lung disease constitutes an enormous group of disorders that lead to significant morbidity and mortality. Our hands remain tight as we investigate understanding ILD in newborns specifically and children, and medical practitioners and families alike are so often confused about describing those conditions and even foreseeing the prognosis and deciding on the management regimens.

In neonates, diagnosis requires more effort as ILD overlaps with similar conditions that present in early infancy (e.g., bronchopulmonary dysplasia, cystic fibrosis, acquired or congenital immunodeficiency, congenital heart disease, bronchopulmonary dysplasia, and pulmonary infection), and thus, a comprehensive history, physical examination, imaging of the chest, bronchoscopy with bronchoalveolar lavage, genetic testing, and finally if the diagnosis is still unclear, lung biopsies, are all needed [1]. The prognosis for children with ILD is wavering, and pulmonary hypertension is what clinicians consider the main predictor of mortality, and children who present with growth failure and fibrosis are deemed to have a poor prognosis [1,2].

Across the literature, the accuracy of ILD as the right terminology to describe these conditions has been questioned, as they are not only limited to the interstitum of the lung, but also expand to involve the alveolar and airway compartments [1], and have even gone further to suggest diffuse lung disease (DLD) as a better definition [3].

The incidence of interstitial lung diseases in children has a range from 0.13 to 16 cases per one hundred thousand of the population [4]. About 50% of cases of interstitial lung diseases of children are detected during the first years of life, but can be diagnosed at any age period, including the adolescence period [2,5]. The incidence of ILD in neonates is unknown due to the scarcity of data.

The recently published remarkable study on acute idiopathic pulmonary hemorrhage in infants (AIPHI), has shed the lights on the idiopathic pattern of lung pathology, and suggested diagnostic criteria that consists of postmortem macroscopic and histological assessment for AIPHI, as clinical assessment is not enough to explain the pattern of the disease [6]. Tachypnea, cough, hypoxemia, crackles, failure to thrive, lung imaging irregularities, and impaired lung function, although they are not solely pathognomonic of pulmonary disease, they are among the common clinical findings that are seen in ILD in children [7,8].

The clinical picture of interstitial lung disease in young children shows the predominance of respiratory disorders with retractions, shortness of breath, and cyanosis. A characteristic auscultatory symptom is diffuse crepitus at the end of inspiration, reminiscent of velcro-type crackles. The first step in diagnosing interstitial lung disease is usually with a chest X-ray. A typical roentgen sign at the early stages of the disease is an increase and deformation of the pulmonary pattern, and a decrease in the transparency of the pulmonary fields by the type of ground-glass opacity, similar to the X-ray pattern of respiratory distress syndrome in premature infants. Young children may lag in physical development and lose weight as a result [9,10].

ILD classification in adults is based on the ongoing histopathology and has generally been classified in adults based upon the underlying histopathology, but the conditions that are observed in adults are different from those of children, and even more distinct in newborns [3].

Collaborative efforts have been established by The National Institute of Health (NIH)-sponsored Rare Lung Disease Consortium (RLDC) and a European Respiratory Society (ERS) Task Force on chronic interstitial lung disease in children to disclose this issue. Their efforts have led to the development of novel concepts and approaches to the assessment of ILD in children. However, the downside of their approach was the unavailability of histopathology for the subjects that were retrospectively studied to lead to their classification [11].

Continuous efforts lead by clinicians, pathologists, and imagers over the past decade have yielded more suitable classifications for ILD in neonates and children based on lung pathology and tend to believe that genetic causes are more likely to be the underlying causative factor for many ILD disorders observed in this age group. However, again, it was based only on children for whom their diagnosis was previously confirmed by lung biopsies, and thus giving less guidance on other cases for whom lung biopsy was not feasible [2]. Another team took it from where this classification ended, and added disorders that were not initially covered that a great number of clinicians would not consider as ILD, such as obliterative bronchiolitis and bronchopulmonary dysplasia (BPD) [12]. Ultimately, this is not the end of the road, and further approaches are required to classify and understand these conditions fully and satisfactorily.

## 2. Relevant Disorders in Infants Aged from 0 to 2 Years Old

### 2.1. Diffuse Developmental Disorders

These are disorders that cause interruption of the development of the lung tissue that ends in diffuse lung disease in term neonates. They are clinically characterized by severe hypoxemic respiratory failure that is irresponsive to treatment. The conditions include congenital alveolar dysplasia, alveolar capillary dysplasia with misalignment of the pulmonary veins (ACDMPV), and acinar dysplasia (also known as type 0 congenital pulmonary airway malformation CPAM) [13].

### 2.2. Alveolar Growth Disorders

These are the most common forms of ILD observed in infants [2,4,14]. The pathology in these disorders is consistent with alveolar simplification [2]. Chest radiography abnormalities include cyst and hyperlucent areas with hyperexpansion, and on computed tomography (CT), ground-glass and linear opacities and subpleural cysts are among the abnormalities observed [15]. Mortalities in these infants is about 34% and predicted by prematurity and growth abnormality [2].

### 2.3. Surfactant Disorders

To date, four genetic mutations are known to be causative of these disorders—SFTPB, SFTPC, ABCA3, and NKX2-1. These genes encode proteins that are required for the production, function, and metabolism of lung surfactant. Infants with these disorders present with radiographically and clinically mimic respiratory distress syndrome in premature neonates [3].

### 2.4. Pulmonary Interstitial Glycogenosis (PIG)

This is a rare disorder that presents in early infancy with diffuse interstitial infiltrates and respiratory distress [16,17]. Interstitial thickening by mesenchymal cells carrying glycogen without fibrosis nor inflammation is the histological presentation of PIG. In imaging, PIG could resemble as diffuse PIG when presented as an isolated condition or patchy PIG, or when it is associated with other lung maturity disorders [18,19]. In CT, PIG features ground-glass opacities, hyperinflated areas, and reticular changes in the subpleural area [14].

### 2.5. Neuroendocrine Cell Hyperplasia of Infancy (NEHI)

This is a disorder that usually presents in early infancy and even the neonatal period with unknown etiology. NEHI is also termed persistent tachypnea of infancy (PTI). It manifests with hypoxemia, retractions, chronic tachypnea, and crackles [20]. Imaging a patient with this condition displays hyperinflation, with high-resolution CT showing the affection of at least of four lobes with air trapping in a mosaic pattern, and GGO distinct to the lingula and right middle lobe [21]. Genetic hypotheses in the development of NEHI is considered after reported familial cases, but no mutations have been linked to the condition as of yet [22]. Prognosis of NEHI is exceptionally good, with most cases recovering gradually, although, there are reports of persistent airway obstruction that resembles severe asthma [23].

## 3. Rare Disorders in Infants Aged from 0 to 2 Years Old

### 3.1. Idiopathic Fibrosing Alveolitis

Idiopathic fibrosing alveolitis is a scarce diffuse lung condition that has two characteristic histological features: the presence of substantial mononuclear cells in the alveolar space, and alveolar wall thickening. Infants with this condition (which is exceedingly rare in infancy, and requires a lung biopsy for confirmation) show symptoms of dyspnea or tachypnea, poor weight gain, cough, and cyanosis. This condition was considered to be an adult-only condition by many clinicians [24], but it can also be observed (although rarely) in infants, usually with with acute illness, and if not properly treated, may result in high mortality and morbidity [7].

### 3.2. Desquamative Interstitial Pneumonitis (DIP)

DIP is diffuse fibrosis of the lung, accompanied by loss of the desquamative functions, which might occur as a nonspecific reaction to different forms of lung injury. It is rare in infants, and its response to treatment with steroids in adults while being irresponsive in children has been reported. In this condition, large alveolar cells show extensive desquamation, while the alveoli show mixed inflammatory exudates usually containing foamy cytoplasm, and macrophages can also be seen [9,10].

Here we present two of extremely rare cases of idiopathic fibrosing alveolitis and desquamative interstitial pneumonitis in two neonates. Written informed consent was collected from the caregivers of our two patients (their mothers). Kazan Medical State University and the Republican Children’s Clinical Hospital Medical Ethics Committee approved this study.

## 4. Cases Presentation

### 4.1. Case 1:

A male neonate born at 37 weeks of gestational age (GA) was transferred to our neonatal intensive care unit (NICU) on his first day of life (DOL) for the evaluation and management of respiratory disorders that appeared within 6 h of delivery.

This patient was born by cesarean section, indicated by a uterine scar from the previous delivery, with a birth weight of 3140 g, 51 cm in length, 34 cm head circumference, and 34 cm chest circumference. He is the second to a healthy sibling. Maternal history was remarked with early hyperemesis gravidarum, pathological weight gain, vaginal candidiasis, and chronic pyelonephritis.

After delivery, the Apgar score was ten at 1 and 5 min of life. At the initial examination, the condition of the infant was assessed as satisfactory, the cry was loud, the skin was pink, the respiration was puerile, and no rales were heard. However, six hours after birth, the condition of the infant started to deteriorate, as he began to grunt, and with auscultation, his breathing was found to have moderately weakened, and while there were no rales, his respiratory rate was up to 66 breathes per minute. There were episodes of cyanosis with a drop in oxygen saturation to 87%, and the infant was tending to be hypothermic.

At this point, the infant was diagnosed with transient tachypnea, and mechanical ventilation was started. For regular sepsis workup, blood analysis and cultures were requested. The white cell count was 13.7/mm^3^ with 38% neutrophils, hemoglobin of 16.8 g/dL, and C-reactive protein (CRP) of 3 mg/dL (normal value for age < 1 mg/dL), respectively. The chest X-ray revealed uneven perihilar infiltrates and patchy infiltrates with a moderate decrease in the lower medial lobe of the right lung with air bronchogram, mediastinum, and sinuses being normal, and no focal shadows were seen. (Figure 1).

On DOL 2, abdominal and cranial ultrasounds were found to be normal, and the cardiac ultrasound revealed patent foramen ovale and closed ductus arteriosus. Nasopharyngeal culture did not show any growth. Despite the efforts of mechanical ventilation, the oxygen saturation continued to drop to 60–70%, on auscultation, and crackles were heard over lower lobes.

By DOL 5, the condition of the infant continued to deteriorate, with aggravation of respiratory failure symptoms and increased dependence on oxygen. Chest X-ray showed air bronchogram and significant reduction of the lung fields (Figure 2), which required endotracheal surfactant administration. CRP raised to 13.2 mg/dL, and white cell count was 14.8/mm^3^ with 84% neutrophils, respectively. Antibacterial and immunoglobulin replacement therapy was subsequently initiated. Cardiac ultrasound conclusion revealed pulmonary hypertension and moderate tricuspid regurgitation. Intensive therapy using the vasodilators and inotropic support was then requested.

On DOL 10, despite mechanical ventilation and intensive therapy, oxygen desaturation persisted, with the chest X-ray revealing bilateral aerial bronchogram and areas of enlightenment in the upper lung fields. (Figure 3).

On DOL 13, respiratory failure was further aggravated, metabolic acidosis progressed, and gastric and pulmonary hemorrhage developed, which required administration of hemostatic drugs.

On DOL 14, the infant succumbed to death, with the autopsy examination revealing a slight decrease in the lungs (to 62 g in weight), a decrease airiness of the pulmonary tissues, and the thickening of the alveolar walls in both lungs. Histological examination revealed peribronchial fibrosis, a sharp thickening and fibrosis of the interalveolar septa, desquamated alveolocytes in all fields, and the development of focal pneumofibrosis.

Pathological diagnosis: Fibrosing alveolitis and cerebral edema.

A flowchart of the progression of our case condition was summarized in Table 1.

### 4.2. Case 2:

A male neonate born at 37 weeks of gestational age (GA) was transferred to our neonatal intensive care unit (NICU) on his first day of life (DOL) for evaluation and management of respiratory disorders that appeared within just 15 min of delivery.

He was born by cesarean section, indicated by the transverse position of the fetus and the premature rupture of membranes, with a birth weight of 3500 g, and 54 cm in length. He is the fifth child to four healthy siblings. Maternal history was remarked with colpitis, anemia, and bacterial vaginosis, with late (at 28 weeks of gestation) registration in the antenatal clinic. There is a history of two abortions.

After delivery, the Apgar score was seven at 1 and 5 min of life, and 15 min after birth, the infant had moaning breathing, with tachypnea up to 64/min, and an oxygen saturation of 86%. The infant was put on continuous positive airway pressure (CPAP), with settings of 30% oxygen and a peep of 6 mbar. The preliminary diagnosis at this stage was mild asphyxia and respiratory disorder of unspecified etiology. Fluid and electrolyte infusion therapy and adapted milk formula were subsequently initiated. A short-term (two hours) improvement was noticed upon therapy, followed by deterioration of his respiratory distress that required mechanical ventilation after eight hours of life. Arterial blood gas (ABG) of the infant showed: pH = 7.48, paO2 = 30 mmHg, and paCO2 = 22 mmHg. Despite the continuous efforts of correction of mechanical ventilation parameters, the oxygen saturation did not improve. X-ray of the chest showed a significant reduction of the lung fields, while blood analyzes and culture results did not reveal any signs of infection or any other abnormalities.

On DOL 2, abdominal, cardiac, and cranial ultrasounds did not show any abnormalities. Nasopharyngeal culture did not show any growth. Despite efforts of mechanical ventilation, oxygen saturation continued to drop to 80%, and with auscultation, moist crackles were heard over the lower lobes bilaterally.

On DOL 3, DOL 5, X-ray of the chest continued to show significant reduction in the lung fields, increased indistinctiveness in the pulmonary pattern, and a decrease in the transparency of the pulmonary fields by GGO. Signs of central cyanosis and hypotension started to develop, and the infant was progressively developing cardio-respiratory failure, which required administration of cardiotonic therapy (dopamine and adrenaline) and enhancement of the mechanical ventilation parameters. The infant died on DOL 5 due to respiratory and cardiovascular insufficiency complicated by pulmonary hypertension. Lung biopsy results showed desquamative interstitial pneumonitis.

## 5. Discussion

In this study, we briefly reviewed ILD in infants under two years of age and present two rare cases of fibrosing alveolitis and desquamative interstitial pneumonitis.

Research on ILD in infants over the past decades has significantly evolved. Despite this fact, a consensus classification of this condition in neonates is still an area of debate. Recent articles have advocated for limiting the fibrosing alveolitis and DIP descriptions only to the adult population who suffer these conditions and have questioned the accuracy of these definitions to describe children and infant patients [11]. Although many studies have included those terms to describe child patients, Refs. [25,26] some studies have gone even further to compare the fibrosing alveolitis and DIP mortality rates between adults and children [27,28,29].

Historically, Scadding (1964) proposed fibrosing alveolitis as a description of the condition when he noticed two cardinal histological features: ① the presence of mononuclear cells in the alveolar spaces, and ② the thickening of the alveolar walls, which was followed by Liebow (1965) observing desquamation of large mononuclear cells in the intra-alveolar space and calling the condition DIP. Thus, these histopathological findings were the main diagnostic features of these conditions [29].

The tendency of considering fibrosing alveolitis and DIP as an adult-only ILD could be based on understanding the role of progressive inflammation as a precursor of fibrous remodeling of the alveolar tissues, but recent studies have established that although inflammation is an important factor, it is not the only culprit of fibrosis formation, and fibrosis is possible to occur even in the absence of inflammation and all of the cases with these conditions do not have an apparent known etiology [5].

It is worth mentioning that the mortality rate in infants with fibrosing alveolitis and DIP is remarkably high (100% in our cases and clinical practice), and they are refractory to treatment. Therefore, further research and cooperation are required to unravel the mystery of these conditions in infants [9].

## 6. Conclusions

The heterogeneity of ILD conditions, the scarcity of resolute research to its prevalence in newborns, and the absence of guidelines on its classification and diagnosis in neonates makes it one of the challenging conditions for pediatricians and neonatologists. Although huge efforts have been made, a universal consensus on classification remains a mission that requires collaborative efforts and ongoing research. Risk factors and underlying causes are almost unavailable, prenatal and maternal conditions, including but not limited to hyperemesis gravidarum, pathological weight gain, gynecological and urinary tract infections, and premature rupture of the membranes could all play a role in initiating such conditions if congenital familial causes are not present (based on our medical practice). Clinical practice, lung biopsy, and histopathological findings remain essential in confirming the diagnosis, especially in rare conditions. Morbidity and mortality rates of ILD in neonates are high, and the mortality rate can even be 100% in rare conditions including fibrosing alveolitis and DIP, since most of these cases are refractory to treatment, especially in newborns. Although genetic testing is becoming more available, results might take several weeks, and with this being mentioned, while time is not always standing by our side when managing such conditions, it could be helpful in counseling for parents when inheritable patterns could be suspected. Although the clinical and diagnostic picture in IDL in neonates is not yet clear, continuous efforts and dedication will untangle this ambiguous condition, paving the road to a bright future.

## Figures and Tables

**Figure 1 children-10-00916-f001:**
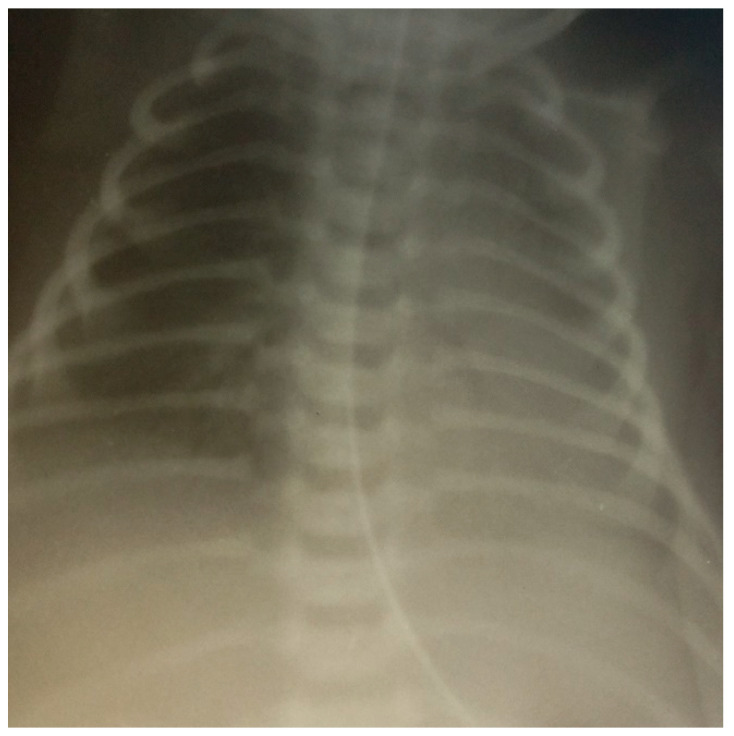
Uneven pneumatic pulmonary fields with a moderate decrease in the lower medial lobe of the right lung with air bronchogram.

**Figure 2 children-10-00916-f002:**
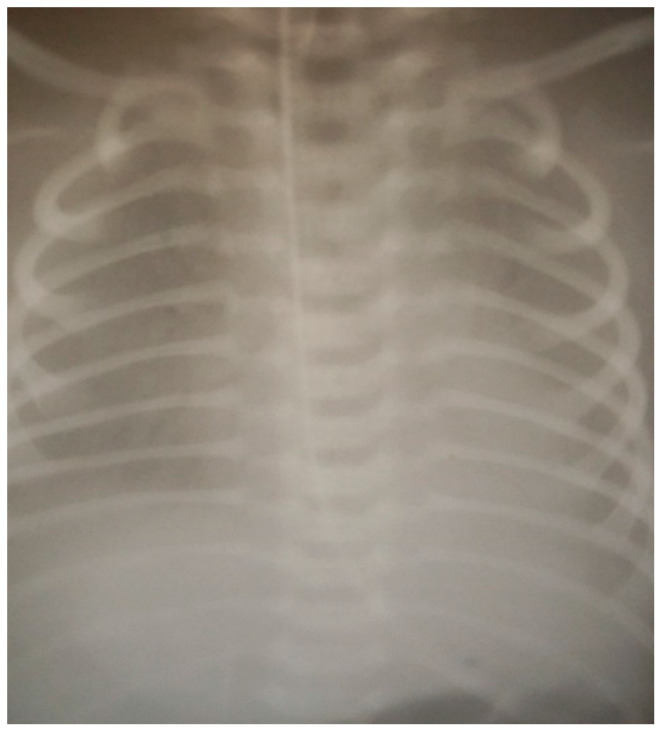
Significant reduction visible in the lung fields.

**Figure 3 children-10-00916-f003:**
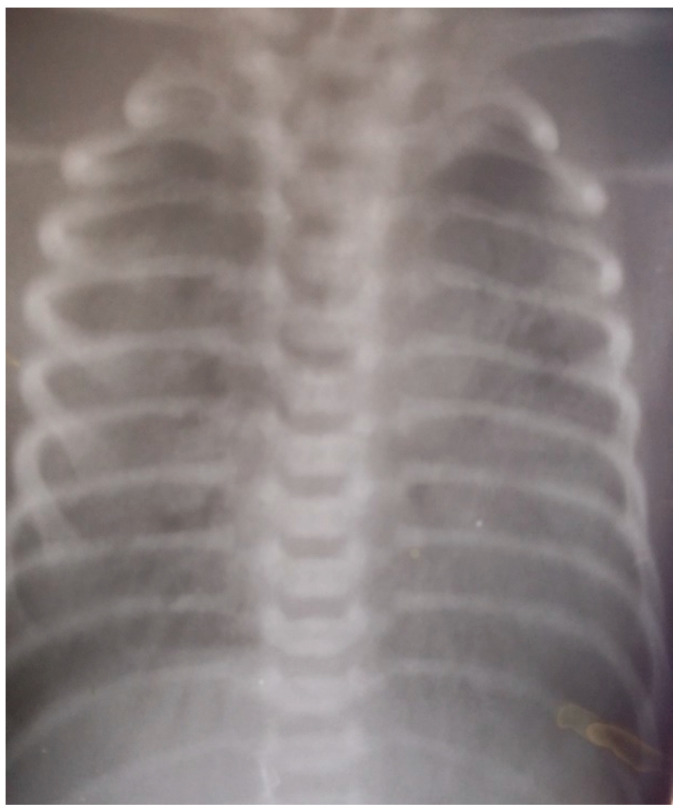
Aerial bronchogram and areas of enlightenment in the upper lung fields on both sides.

**Table 1 children-10-00916-t001:** A Flowchart of the progression of the condition of the first case.

	Day of Life (DOL)					
	DOL 1	DOL 2	DOL 5	DOL 10	DOL13	DOL 14
Oxygen saturation	86%	60–70%	60–70%	60–70%	60–70%	
Cardiac U/S		Patent foramen ovale and closed ductus arteriosus	Pulmonary hypertension, moderate tricuspid regurgitation			
Cranial U/S		Normal				Cerebral edema
Abdominal U/S		Normal				
CRP	3 mg/dL		13.2 mg/dL			
WBCs	13.7/mm^3^, 38% neutrophils		14.8/mm^3^, 84% neutrophils			
X-ray	See Figure 1		See Figure 2	See Figure 3		
Lung biopsy	N/A	N/A	N/A	N/A	N/A	Fibrosing alveolitis

U/S, ultrasound; CRP, C-reactive protein; WBCs, white blood cells; and N/A, not applicable.

## Data Availability

All data and materials for this study are available upon request by the Editor-in-Chief of this journal.

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
