# Peer review of "Interstitial Lung Disease in Neonates: A Long Road Is Being Paved"

_children, 2023, doi:10.3390/children10060916_

Round 1

Reviewer 1 Report

Line 62. these types of symptoms are not pathognomonic of pulmonary disease. I think you need to stress the idiopathic nature of lung patterns in newborns. you can also quote this: doi:10.3390/diagnostics13071270. 

you have done histological sample. Could you provide photos? at line 207 for example, you wrote: "Histological examination revealed peribronchial fibrosis, a sharp thickening and fibrosis of the interalveolar septa, desquamated alveolocytes in all fields, and the development of focal pneumofibrosis". Here, this is a chronic disease, not an acute one. Do you think that pathological process began in the intrauterine period? It could be conseguent to a tricuspid regurgitation.

Conclusion: where is the news? there are any alternative approach do you advise? Consider to wait and show genetic tests.

Author Response

Reviewer 1:

  • Line 62. these types of symptoms are not pathognomonic of pulmonary disease. I think you need to stress the idiopathic nature of lung patterns in newborns. you can also quote this: doi:10.3390/diagnostics13071270. 

Response:

Thank you so much, we have rewritten those symptoms and made it clear to understand and have cited the recommended article and we have found it very interesting and very helpful.

  • you have done histological sample. Could you provide photos? at line 207 for example, you wrote: "Histological examination revealed peribronchial fibrosis, a sharp thickening and fibrosis of the interalveolar septa, desquamated alveolocytes in all fields, and the development of focal pneumofibrosis". Here, this is a chronic disease, not an acute one. Do you think that pathological process began in the intrauterine period? It could be conseguent to a tricuspid regurgitation.

Response:

Thank you so much for this interesting point, unfortunately, we accidentally discarded those photos, since we have done this work over three years ago, but we still have the reports officially signed and stamped.

The debate and the novelty of our study are exactly what you are questioning here, how come this chronic pattern of the disease appears in neonates? Since the diagnosis was confirmed by a histological exam, the pathological process must have begun in the intrauterine period. We really appreciate that you share the same conclusion with us.

  • Conclusion: where is the news? there are any alternative approach do you advise? Consider to wait and show genetic tests.

Response:

Thank you so much for mentioning this point. Unfortunately, as the wonderful article on AIPHI you suggested to quote has pointed the mortality rates in diseases of this nature are really high and they require a post-mortem macroscopic and histological assessment, to the best of our knowledge, the idiopathic nature of ILD had made it difficult to manage and predict. Here, we advise genetic counseling for parents and genetic testing to determine the responsible gene/ genes. Techniques such as mitochondrial donation treatment (per se) would be a solution for such families, where inheritable patterns could be suspected to prevent neonates from inheriting incurable diseases.

 We have added this point to our conclusion. Thank you again.

Reviewer 2 Report

The present article deals with two ILD cases in neonates. I admit the topic is very interesting and gathering information is welcomed and could prove to be very helpful for clinicians. However, I have two major general comments:

General comment 1: It would be very enlightening if the authors could provide pathology images of the biopsy sections of the two cases

General comment 2: I think that the authors should provide more information concerning the therapeutic strategy followed in each case

Moderate editing of English language is in my opinion required. For example:

Lines 19-21: "Although adult and children classifications for ILD have evolved over the years, classification for ILD in neonates is still a challenge". This is mentioned twice. Please subtract one of the two identical sentences.

Lines 24-25: Please consider to rephrase. Do you mean that the ILDs diagnosed in two neonates were previously considered as a diagnosis only in adults?

Author Response

Reviewer 2:

The present article deals with two ILD cases in neonates. I admit the topic is very interesting and gathering information is welcomed and could prove to be very helpful for clinicians. However, I have two major general comments:

  • General comment 1: It would be very enlightening if the authors could provide pathology images of the biopsy sections of the two cases

Response:

Thank you so much for this interesting point, unfortunately, we accidentally discarded those photos, since we have done this work over three years ago, but we still have the reports officially signed and stamped.

We agree upon that postmortem macroscopic and histological assessment is fundamental to reach the diagnosis.

  • General comment 2: I think that the authors should provide more information concerning the therapeutic strategy followed in each case

Response:

Thank you so much for mentioning this point. Morbidity and mortality rates of ILD in neonates are relatively high and the mortality rate could be 100% in some rare conditions like fibrosing alveolitis. Managing the complications and the accompanying pathological conditions, such as pulmonary hypertension, pulmonary hemorrhage, and cardiovascular insufficiency is required, but does not decrease the mortality rates.  The idiopathic nature of ILD had made it difficult to manage and predict.

Here, we encourage genetic studies on such cases to determine the causative gene/ genes and help counseling for parents when inheritable patterns could be suspected.

  • Comments on the Quality of English Language

Response:

Thank you so much for this suggestion, we have edited what was suggested. Really appreciate your time and suggestions.

Moderate editing of the English language is in my opinion required. For example:

Lines 19-21: "Although adult and children classifications for ILD have evolved over the years, classification for ILD in neonates is still a challenge". This is mentioned twice. Please subtract one of the two identical sentences.

Response:

Thank you so much for this suggestion, we have edited that.

Lines 24-25: Please consider to rephrase. Do you mean that the ILDs diagnosed in two neonates were previously considered as a diagnosis only in adults?

Response:

Thank you so much for this suggestion, we have edited that to make it clear to understand.

These two diagnoses have a chronic pattern, which points out that they might start during the fetal period. Some work of literature considers Idiopathic Fibrosing Alveolitis and Desquamative interstitial pneumonitis diseases of adulthood.

Round 2

Reviewer 2 Report

The authors  have addressed all my comments